# Strategic human resource management practitioners' emotional intelligence and affective organizational commitment in higher education institutions in Georgia during post-COVID-19

Roya Anvari[1]*, Vilmantė Kumpikaitė-Valiūnienė[2], Rokhsareh Mobarhan[3], Mariam Janjaria[1], Siavash Hosseinpour Chermahini[4]

1 School of Business and Administrative Studies, The University of Georgia, Tbilisi, Georgia, 2 School of Economics and Business, Kaunas University of Technology, Kaunas, Lithuania, 3 Danesh Alborz University, Abyek, Qazvin Province, Iran, 4 School of Science and Technology, The University of Georgia, Tbilisi, Georgia

* r.anvari@ug.edu.ge

## Abstract

The COVID-19 pandemic has significantly affected the global workforce, presenting unprecedented challenges to managers and practitioners of strategic human resource management. Pandemic-influenced changes in the employment relationship highlighting the need for adaptation in order to facilitate a return to pre-pandemic conditions. Crises such as this can have a detrimental effect on employees' psychological contract, which in turn can hinder the organization's ability to thrive in the post-COVID-19 era and impede the development of high commitment levels in the aftermath of the crisis. Emotional intelligence plays an increasingly vital role in effectively navigating the crisis and providing support to employees, while also facilitating the reconstruction of the psychological contract. Therefore, this study aims to explain the role of emotional intelligence of strategic human resource management practitioners on affective organizational commitment and the possible mediating effect of the psychological contract in that relationship. A quantitative study took place in February 2023 among 286 HR directors, HR managers, and HR officers in higher education institutions in Georgia. Partial Least Squares for Structural Equation Modelling was applied for data analysis. The results revealed that the emotional intelligence of strategic human resource management practitioners has a positive impact on the psychological contract and the affective organizational commitment. This study supports the idea that emotional intelligence can transform strategic human resource management practitioners into individuals who engage in people-orientated activities. These activities aim to effectively acquire, utilize, and retain employees within an organization. The study also suggests that emotional intelligence can provide solutions to maintain high employee commitment during times of crisis and in the aftermath of unprecedented situations.

**Data Availability Statement:** All relevant data are within the paper and its Supporting Information files.

**Funding:** The authors received no specific funding for this work.

**Competing interests:** The author declares no conflict of interest.

## 1. Introduction

The workplace has experienced unparalleled effects due to COVID-19. This situation has unprecedentedly impacted the workplace, presenting managers and human resource management (HRM) practitioners with a complex and challenging environment. Initially, HR played a crucial role in managing the organizational response to COVID-19, including tasks such as workforce placement, work arrangements, and task completion [1]. The world is trying to recover from the COVID-19 pandemic and changes in employment relationships have already been seen [2, 3]. The current post pandemic situation presents an opportunity for strategic human resource management (SHRM) practitioners to demonstrate self-awareness and resilience in overcoming the effects of COVID-19. By integrating emotional intelligence (EI) competencies into SHRM activities, practitioners can effectively utilize and retain employees [4–6]. For example, the research conducted in Georgia, revealed that individuals with high EI can mitigate the impact of job insecurity on their affective organizational commitment (AOC), which becomes more critical in post-crisis situations. Additionally, Nordin [7] found that individuals with high EI possess better coping strategies compared to those with low EI. Therefore, it could be noted that organizations must not only rely on intelligence but also develop emotional stability to thrive in the future. A review of existing literature shows that numerous studies have already been conducted on SHRM in higher education across different countries [8–11]. First, previous studies have examined SHRM issues such as training, compensation, and performance appraisal, highlighting their significance among administrators and policymakers in Indonesia [11] Spain [9] Russia [12], and Saudi Arabian higher education. Additionally, according to Allui and Sahni [8], these studies underscore the scarcity of data concerning the link between faculty satisfaction and retention, which is a critical issue in higher education.

Second, research on the motivations for HR strategies suggests that problem-solving abilities, perceived assistance, social changes, and study challenges play important roles. Hunter [10] found that concerns about people management in organizations, lack of university autonomy, and the influence of higher education ministries can impact employees' behavior, as seen in employment challenges faced by administrative staff in Malaysia following COVID-19. Bitsadze and Japaridze [13] emphasize the significance of studying SHRM and organizational commitment (OC) in specific contexts, particularly in Georgia. They also highlight the need for future research on strategic decision-making, self-evaluation of research personnel, and commitment to continuous development in Georgian educational institutions to align with European higher education standards [14]. However, it could be revealed a shortage of scholarly investigations exploring the connection between SHRM practitioners' EI and AOC in higher education institutions. Specifically, there is a lack of research addressing the post-COVID-19 environment. In order to bridge this research gap, our study seeks to investigate the impact of EI among SHRM practitioners on AOC, particularly in the context of the post-COVID-19 era. This paper looks at how SHRM practitioners can reimagine their own future in terms of managing their employees. Drawing on insights from psychology, the importance of attending to the human side of employment relationships is emphasized. Therefore, this research seeks to enhance the current body of knowledge on SHRM and the advantages of emotional treatment. The primary objective of this research is to examine how psychological contract (PC) mediate the relationship between the EI of SHRM practitioners and AOC within higher education institutions in post-COVID-19 Georgia. The paper follows a structured format, including a theoretical background, hypotheses, methodology, data collection, results, and practical and theoretical implications for organizations.

## 2. Theoretical framework

### 2.1 EI conceptualization and EI in post-COVID-19 era

The concept of EI has garnered increasing interest from academics in the social sciences since the early 20th century [15]. However, despite this growing interest and its relatively young history, different authors have defined and conceptualized EI in various ways. The foundations of EI can be attributed to the studies conducted by Thorndike in the 1920s [16]. Salovey and Meyer [17] later expanded on the emotional competences outlined by McClelland [18], Gardner's [19] interpersonal intelligence, and Thorndike's intuition to provide further insight and clarification in their scientific study. In this study EI is defined and introduced as the ability to accurately perceive, assess, and express emotions; the capacity to access and create emotions that support cognitive processes; the proficiency in understanding emotions and emotional knowledge; and the ability to manage emotions to promote emotional and intellectual growth [20]. Goleman [21] further described EI competencies as essential skills that cultivate empathy, self-discipline, and teamwork to enhance work performance. The definition of EI is classified into two primary categories: the ability model and the mixed or trait models [22, 23]. The ability model views EI as a cognitive ability involving the use of emotionally charged information [24]. It emphasizes an intellectual understanding of emotions and their influence on thoughts and actions [25]. Mixed or trait models integrate personality dimensions, such as optimism, assertiveness, and empathy, with cognitive factors related to emotion, such as perception, assimilation, understanding, and management. These models have been proposed by researchers like, Bar-On [26], Goleman [21] and Petrides et al. [27]. Joseph and Newman [28] argue that the second model, which combines personality traits with cognitive factors, is a stronger predictor of job performance. The year 2020 brought significant shifts, making EI even more critical for leaders and managers in public and private organizations. Managing employees and maintaining an emotional connection has always been challenging, and the importance of EI continues to grow. In the post-COVID-19 era, the demand for higher EI has increased. Recognizing and understanding emotions, both in oneself and others, is linked to reduced burnout, increased commitment, and proactive coping strategies during and after the pandemic [29]. Managers recognize the importance of EI for generating followership and performance. Employees seek a deeper connection beyond numbers, wanting to be heard, understood, and valued. Understanding why EI is more essential than ever is crucial in navigating the post-COVID-19 world.

### 2.2 Conceptualization of SHRM and COVID-19

Strategic human resource management (SHRM) is a concept that emerged in the 1990s, combining the principles of strategy and human resource management. It recognizes the importance of aligning HRM practices with organizational strategy to achieve a competitive advantage [30]. Today, SHRM is widely recognized as the interaction between human resource management and activities aimed at achieving organizational goals. It involves deploying activities to achieve organizational success through planned behaviors and addressing long-term HR issues as part of strategic management. SHRM encompasses two main perspectives: HRM as an integral part of organizational strategy and HRM as a strategy in itself, focusing on developing staff to navigate a rapidly changing world. During the COVID-19 pandemic, SHRM became even more crucial in managing challenges and adapting work organization and job design [31]. HRM practitioners implemented guidelines and crisis management to support employees, reduce stress, and increase motivation and confidence. Strategic thinking and contemporary HR practices such as flexibility and job design are essential in effectively

managing pandemic-related challenges and aligning HRM with strategic planning [32]. It is consistent with the human capital theory which reflects the strategic significance to human resources. In the post-COVID-19 era, adopting SHRM helps organizations effectively manage challenges and improve job performance [33]. Detailed HRM strategies are necessary to address staffing, training, compensation, and performance appraisal in alignment with strategic planning.

**2.2.1 Staffing in post- COVID-19.** Staffing involves the process of locating, selecting, evaluating, and nurturing employees [34]. The dynamics of staffing have undergone significant changes [35]. The pandemic has disrupted operations, making it difficult for organizations to anticipate their staffing requirements, even in the short term [36]. HR departments must adapt to current trends and carefully assess both immediate and future resource needs [36]. During the pandemic, organizations have commonly implemented strategies such as partially or completely halting recruitment [37]. COVID-19 has influenced various aspects of recruitment, including changes in hiring priorities, employer branding, flexible work arrangements, and telecommuting [37]. The prevalence of remote work options is expected to continue in organizations [38]. In the healthcare sector, there has been an increase in programs and resources for mental health, workshops, and employee resiliency training in post-COVID-19 nurse staffing [39]. Supporting managers and employees throughout the staffing process and providing them with accurate information pose major obstacles for human resource managers during the pandemic. In the post-COVID-19 era, SHRM practitioners play a vital role in addressing staffing challenges by considering the following factors:

- Anticipating managers' expectations regarding staffing changes in the next three to five years (Green, 2022).

- Developing strategies for attracting new talent to the organization.

- Defining the staffing strategy for the post-COVID-19 era and adapting goals since the outbreak of the pandemic.

- Employees' work arrangement preferences in the post-COVID-19 landscape.

- Employees' expectations for different employment classifications (full-time, part-time) in higher education in the post-COVID-19 era (Green, 2022).

In response to the circumstances, organizations have adopted and utilized digital or online approaches for the process of attracting, evaluating, and selecting candidates for job positions which presents significant challenges for HR professionals and those who are actively looking for employment opportunities. However, companies have successfully utilized virtual recruiting practices, resulting in time, cost, and resource savings. Virtual recruitment is anticipated to continue as a standard practice, especially for entry-level positions and initial screening stages in the post-COVID-19 environment. Research conducted by Bieńkowska et al. [36] indicated that human resources people were ill-equipped to handle this unforeseen change in workforce. Hence, it is crucial for SHRM practitioners to develop strategies to retain talent in the post-pandemic period. While COVID-19 is still present, certain recruiting and staffing practices are expected to endure in the post-pandemic world.

**2.2.2 Training in post-COVID-19.** The COVID-19 crisis has brought significant changes to working conditions in organizations. Managers have sought to go beyond traditional training methods and implement strategies to develop the skills required by employees, increase awareness about the new virus, and support remote working [40]. One of the major challenges for HRM practitioners during this crisis has been the development of new training programs that consider physical distancing and the adoption of new training methods in line with the

new reality of employees and organizations. Thilagaraj and Rengaraj [41] highlight that the post-COVID-19 workplace will require different professional techniques compared to the pre-pandemic period. This means that SHRM practitioners need to provide more coaching and their training teams need to be innovative in their approach. SHRM practitioners face challenges and a significant opportunity to revisit traditional learning models and find better ways to support their teams. It is evident that returning to normal is not an option, as we are in a new post-COVID-19 environment. Managers also need to reconsider their training methods to meet the changing needs and uncertainties of the future. Consequently, they are exploring new and innovative ways to deliver accurate, timely, and practical training [41].

In the post-COVID-19 era, SHRM practitioners need to address the following questions to overcome training challenges:

- How can SHRM practitioners address training issues in the post-COVID-19 era?

- How can strategic thinking contribute to the success of training plans in the future?

**2.2.3 Compensation and post-COVID-19.**   The term compensation describes a monetary payment made to a person in return for their services. According to Dessler [42] employee compensation and benefits comprise all forms of pay that employees get that are related to their employment in addition to their regular earnings or salary. They include both direct financial compensation (bonuses, profit-sharing) and indirect financial compensation (medical care, health insurance, paid time off, etc.). Three categories are used by Leibowitz [43] to classify benefits. The first one comes with health insurance and is nontaxable for personal use. There is taxation on the second and third categories. Benefits like life insurance, which the employer can provide at a reduced cost because of quantity discounts, fall under the second category; paid time off falls under the third [43]. The COVID-19 pandemic has not only brought about changes in employee lifestyles, needs, and preferences, but it has also presented unique challenges for companies' executive compensation programs. These challenges include setting goals for incentive plans in an uncertain environment, implementing cyclical incentives, and performance-based grants, providing temporary paid sick leave. To address these challenges, organizations have developed various approaches in compensation strategies and policies as a primary motivator. For example, Arnold and Sirras [40] recommend changing compensation strategies and frameworks to incorporate discretion in assessing performance, avoiding layoffs, and retaining employees or hiring new employees at lower salaries in remote geographic regions. However, despite the strategic importance of compensation and the challenges posed by the new normality, limited attention has been given to this area in higher education [44]. Therefore, it is crucial to rethink the underlying principles for compensation in the post-COVID-19 era. This suggests that there is a chance to make a significant impact on employees' well-being and financial security, even after the COVID-19 pandemic, by reevaluating and adapting compensation and benefits within organizations [38].

**2.2.4 Performance appraisal and post-COVID-19.**   Performance appraisal involves systematically identifying and evaluating employees across various performance dimensions to ensure alignment with organizational goals and value for the organization's investment [45]. It serves as a valuable feedback mechanism for employees and managers and helps managers identify employees with potential for promotion. The COVID-19 pandemic has significantly impacted performance appraisal, presenting challenges such as disruption in performance-based pay, workplace isolation, lack of communication and control, and role conflict [46, 47]. These challenges have made it challenging for SHRM practitioners to assess performance during remote work, as trust has been compromised and tensions between employees and

employers have increased. Aguinis and Burgi-Tian [48] highlight that a lack of communication, data, uncertainty, and feedback can create critical issues in evaluating performance during a crisis. In this context, it is crucial for managers to adapt and conduct fair assessments that provide stability, motivation, and recognition for employees' contributions. Transparency, clarity, and the adoption of innovative and adaptable approaches can help maintain trust in the post-COVID-19 era. Gallup [46] emphasizes the importance of re-engineering performance management to address challenges such as infrequent feedback, lack of clarity, manager bias, and negative reactions to evaluation and feedback during the COVID-19 era. Some key considerations for performance appraisal in the post-COVID-19 workforce include identifying effective methods for appraising performance in the new environment, recognizing employees who went above and beyond during challenging times, evaluating performance in relation to adjusted goals, providing effective feedback, and planning performance appraisals that deviate from the usual method of reviewing individual job performance and overall contribution to company performance [49]. By addressing these questions and adapting performance appraisal practices, organizations can navigate the unique challenges posed by the pandemic and ensure that employees are acknowledged and rewarded for their efforts.

## 2.3 PC conceptualization and PC in the post-COVID-19 era

Rousseau and McLean Parks [50] proposed a model with five dimensions based on Macneil's [51] contractual continuum. These dimensions include time frame, tangibility, scope, focus, and stability. Time frame pertains to how long individuals perceive their employment relationship to last, while tangibility relates to the clarity and specificity of the contract terms. Scope examines the permeability between work and personal life, while focus and stability explore the ability of the PC to evolve and change over time. The PC is rooted in social exchange theory and represents mutual obligations in the employee-employer relationship [52]. It encompasses transactional, relational, balanced, and transitional dimensions, focusing on economic obligations, commitment, communication, stability, and performance. The COVID-19 pandemic has significantly impacted the employee-employer relationship and the PC [53]. Leaders face challenges in navigating the changing working conditions and fulfilling the PC in the post-COVID-19 world [53, 54]. Studies show how the pandemic has influenced the nature of the PC and shifted expectations for both employees and employers [53, 55–58]. SHRM practitioners play a crucial role in implementing strategies to meet the needs of both parties and ensure stable employment relationships [59]. Understanding the impact of the crisis on SHRM and employment relationships is important, as well as recognizing the challenges and unattainable expectations in the post-COVID-19 era [60]. It is necessary to examine how the employee PC has changed and how its fulfillment affects attitudes and behaviors in the new working environments.

## 2.4 AOC conceptualization and AOC in post-COVID-19 era

According to Limon [61] organizational commitments understand and defines their shared values, beliefs, norms, and behaviors is known as the conceptualization of OC. Three aspects of organizational commitment were recognized by Meyer et al. [62] as affective, continuance, and normative commitment. Normative commitment stems from a sense of duty, continuance commitment arises from the need to continue working, and affective commitment is a sincere wish to stay with the company. Historically, OC has been perceived as a stable and enduring aspect of organizations that shapes their identity and influences employee conduct [63]. However, in the post-COVID-19 era, there has been a shift in the conceptualization of OC to recognize the dynamic and adaptive nature of organizations. The pandemic has compelled

organizations to swiftly adapt to remote work, digitalization, and evolving business models. Consequently, OC is now being seen as more flexible, resilient, and responsive to external changes [63]. Organizations are increasingly aware of the need to foster commitment that promotes agility, innovation, and employee well-being to navigate the uncertainties of the post-COVID-19 world. According to Becker [64], one perspective on commitment is based on the exchange between individuals and organizations, where a more favorable exchange leads to greater commitment. It reflects loyalty and performance, with higher commitment associated with increased loyalty which encompasses a sense of belonging and citizenship behaviors that enhance efficiency [65]. During the COVID-19 pandemic, employee commitment has been a challenge for organizations [63, 66, 67]. Akartuna and Serin [63] found that teachers' commitment decreased due to changes in working conditions. The importance of increasing emotional attachment and involvement in the organization was emphasized to maintain commitment and prevent turnover intentions during the pandemic [66]. Low levels of AOC were observed among both female and male teachers. Moving forward from the crisis, SHRM practitioners have an opportunity to develop strategies that foster positive attitudes, enhance employee engagement, and drive organizational success in the post-COVID-19 era. Therefore, AOC which pertains to emotional attachment is the focus of this study.

## 3. Hypotheses

### 3.1 The SHRM practitioner's EI and AOC

Within the theoretical framework outlined earlier, it is essential to further elucidate the mechanism through which HRM strategies impact organizational commitment. SHRM practitioners are responsible for deploying activities and creating a work atmosphere that fosters inclusion and acceptance and devotion to the workplace, ultimately enabling companies to achieve their goals [68, 69]. Management strategies can influence AOC by fostering a positive sense of responsibility and emotional attachment among employees. Reviewing the literature, it is argued that HRM practices and the employment relationship play a vital role in increasing employee commitment [36].

EI is crucial for SHRM practitioners, especially in a post-COVID-19 environment. Asserted by Goodlet et al. [70], developing EI allows leaders to consider employees' attitudes, understand their needs, and contribute to organizational effectiveness through effective relationship management. Research shows that team leaders with high EI can positively influence employees' commitment and proactive behavior, particularly during stressful situations like the COVID-19 outbreak [70]. Alternatively, employees who have a diminished capacity for EI often find themselves grappling with increased levels of anxiety as well as burnout [71]. Therefore, based on Zampetakis and Moustakis's research [72] it is important for SHRM practitioners to effectively manage their own emotions to avoid negative impacts on their actions. Effective HRM is rooted in EI, as it helps managers perform better, be more productive, and reduce conflicts with others [5]. Leaders with high EI excel in communication, conflict management, and guiding employee performance [21]. Emotional competences, such as developing others, understanding needs, and effective communication, are essential for SHRM practitioners to understand employee needs, provide a sense of security, and maintain faculty and administrative vitality. Positive emotions at work, such as inspiration, happiness, and enthusiasm, influence rational reductions in costs related to training, development, performance appraisal, retention plans, and compensation [63], which are important challenges in the post-COVID-19 era. The organization's response to the COVID-19 outbreak can influence changes in employees' affective commitment, and leaders' EI is crucial in fostering energetic motivation and happiness among employees [6]. When examining the three aspects of

organizational commitment (affective, continuance, and normative), it becomes apparent that the EI of SHRM practitioners is strongly associated with affective commitment, which pertains to emotional attachment. Thus, emotionally intelligent SHRM practitioners who employ practices such as staffing, training, compensation, and performance appraisal are more prone to embrace strategies that influences AOC and a sense of belonging in the higher education sector, especially in the face of the COVID-19 crisis. This is particularly important in the post-COVID-19 environment. Therefore, we can state the hypotheses as follows:

**H1**. EI of (SHRM) practitioners has a positive effect on their SHRM practices in the higher education in post-COVID-19.

**H2.** EI of (SHRM) practitioners has a positive effect on their AOC in the higher education in post-COVID-19.

### 3.2 The SHRM practitioner's EI and PC

The literature highlights the significance of HR for organizational strength. Organizations, based on resource-based theory, select qualified employees who contribute to organizational development [73]. Social exchange theory supports the idea that motivation shapes positive relationships [74]. Research on HRM strategies and the PC shows that effective SHRM optimizes skills, fosters support, and strengthens the PC [73, 75–77]. Supporting employees during the COVID-19 outbreak significantly impacts their feelings and reciprocal relationships [78]. Building positive emotions in the workplace is crucial for employees' well-being [79]. Managers' EI affects the PC, as low EI leads to ineffective HRM practices and negative attitudes [80]. The COVID-19 pandemic has disrupted employment relationships, causing shifts and breaches in psychological contracts [56]. SHRM practitioners play a vital role in ensuring stable employment relationships and addressing challenges in the post-COVID-19 era [59]. Understanding and negotiating the PC during times of stress according to Francesca Di Meglio [60], particularly in higher education, is important. Therefore, it can be hypothesized that:

**H3:** EI of (SHRM) practitioners has a positive effect on PC in the higher education in post-COVID-19.

### 3.3 PC and AOC

Existing research has primarily focused on the PC, examining content-based elements (transactional and relational) and evaluation-based elements (fulfillment or breach) [81–83]. These studies have explored the impact of psychological contracts on work-related attitudes like job engagement, job involvement, and OC [57, 84–87]. Coyle-Shapiro and Kessler [88] found that PC fulfillment directly influences employees' affective commitment. Yu et al. [57] demonstrated the positive impact of PC fulfillment on intrinsic motivation and affective commitment during the COVID-19 outbreak. Akartuna and Serin [63] showed changes in teachers' commitment levels due to the pandemic. However, the effects of these extraordinary circumstances on work-related attitudes and behaviors in the post-COVID-19 era are still unknown [57, 89]. Based on these arguments, we propose that psychological contracts positively affect AOC, which aligns with attitudes in the post-COVID-19 era. Therefore, we hypothesize that:

**H4.** PC of (SHRM) practitioners has a positive effect on their AOC in the higher education in post-COVID-19.

### 3.4 PC mediates association between SHRM practitioner's EI and AOC in post-COVID-19

Based on the arguments presented, it is suggested that a mediation model can better explain the relationships. This means that the EI of SHRM practitioners can impact on AOC either directly or indirectly. Indirect impact occurs through the influence of SHRM practitioners' EI on the establishment of PC, which then motivates employees to exceed their job descriptions and align with organizational goals, leading to increased AOC. In the post-COVID-19 era, when uncertainty is prevalent, the importance of EI as a core HR competency is even higher [90]. Taking into account the relationships discussed, we can propose a hypothesis for mediation. Firstly, previous studies have demonstrated that psychological contracts have a significant impact on organizational commitment, indicating a clear connection between the two variables [84, 86]. PC fulfillment has been found to significantly affect employees' emotional attachment during the COVID-19 crisis [57], while breach of the PC can lead to absenteeism and turnover. Secondly, SHRM practices influence employee attitudes and behaviors, including AOC. These practices foster positive employee attitudes towards the organization and elicit positive responses based on reciprocity, particularly during the COVID-19 outbreak when employee performance is crucial [36, 91]. SHRM, as a modern approach, involves all management levels in promoting AOC and motivation in today's organizational context [92]. Given the theoretical background, it is important to examine the mechanism of SHRM practitioner influence on AOC while considering the mediating role of psychological contract. Furthermore, in the post-COVID-19 era, organizations are faced with the challenge of establishing a happy and engaged work environment, where there is a greater demand for EI. Incorporating EI into SHRM practices, including training, compensation, staffing, and performance appraisal, can empower organizations. In summary, it is argued that the EI of SHRM practitioners and PC in the post-COVID-19 era can enhance employees' affective commitment to the organization.

**H5:** PC mediates the positive relationship between SHRM practitioner's EI and AOC in the higher education in post-COVID-19

### 3.5 Final model of hypotheses

The hypotheses outlined in Fig 1 had been examined through empirical research. This research involved measuring each variable and utilizing statistical analysis to identify the relationships between them. By conducting the research in this manner, we will be able to gather evidence and determine whether the hypotheses are supported or refuted. This approach will contribute to our understanding of the variables and their connections.

## 4. Methodology

This section contains (i), design of the research, (ii) the instrument, (iii) the sample technique, and (iv) the data collection. In this study, a quantitative approach is utilized to investigate the connection between the EI of SHRM practitioners as well as AOC in the educational sector, specifically among administrators and policy makers. The study follows a deductive approach and employs a cross-sectional design. The main objective is to examine how the PC mediates the impact of SHRM practitioner's EI on AOC in the post-COVID-19 environment. This research primarily relies on data collected through literature reviews. This includes reviewing books and journals from libraries, as well as conducting internet research to explore the perspectives of other scholars. For the selection of participants, HR directors, HR managers, and HR officers were chosen using a purposive sampling technique based on availability. A total of

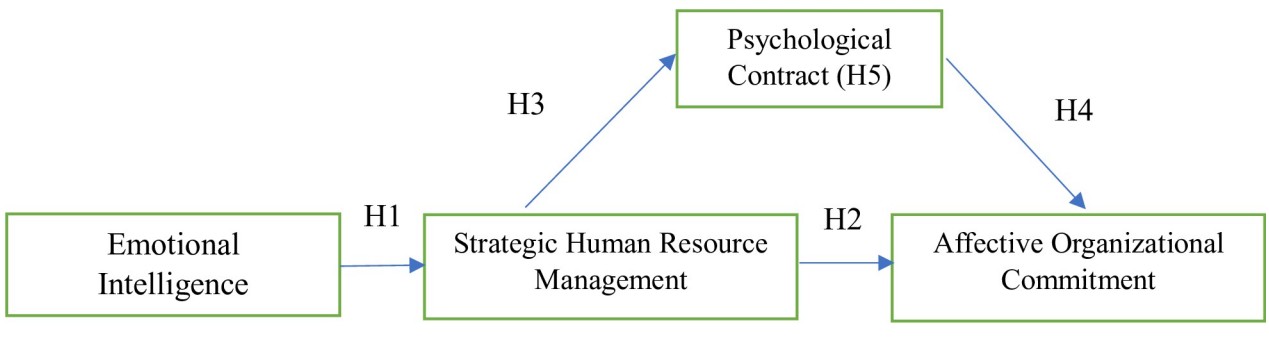

**Fig 1. Conceptual framework.**

393 HR directors, HR managers, and HR officers were selected from universities in the zone. In Georgia's higher education industry, there are 63 higher education institutions, including 34 universities, 23 teaching universities, and 6 colleges in 2022. This study specifically focuses on Tbilisi, the capital of Georgia, which is home to the largest number of public (12) and private universities (30). In Georgia, each university has between 5 to 20 managers responsible for human resources. The target sample for this study encompasses all these managers in the 42 public and private universities in Tbilisi, Georgia, amounting to approximately 393 individuals. To achieve a level of 99% confidence with a confidence interval of 5%, the study would require 251 respondents. Alternatively, a confidence level of 95% with a margin of error of 5% would require 197 respondents, as calculated using the Survey Monkey calculator. Data collection took place from February 2023 to March 2023, using an online questionnaire. The survey was shared with all members of a WhatsApp group consisting of HR managers from universities in Tbilisi. In the data collection phase, a grand total of 286 participants from both public and private universities completed the questionnaire. IBM SPSS software was utilized for analyzing data and Partial Least Squares (PLS) for Structural Equation Modeling (SEM). PLS-SEM was chosen for its capability to assess both direct and indirect relationships among variables. SEM allows for the examination of structural models, which reveal the dependencies between exogenous variables (independent, unexplained, or not predicted by other variables) and endogenous variables (dependent variables).

## 4.1 Variables and measures

All constructs were measured using established scales, with some modifications made to better capture the post-COVID-19 context, which was the focus of this study. The survey comprised four sections, including a total of 56 items. The initial section measured practitioners' EI with 16 items, the second part assessed SHRM practices with 17 items, the third part evaluated the PC with 17 items, and the fourth part measured AOC with 6 items. All survey items were assessed using a Likert-scale, comprising five response choices ranging from 1 (strongly disagree) to 5 (strongly agree).

**4.1.1 Emotional intelligence.**   The measurement of EI in this study followed the approach of Molina et al. [6]. Their self-report scale, which is suitable for HRM research and easily understandable by respondents, was utilized. Self-report measures are effective in assessing individuals' emotional abilities [93]. The scale consisted of 16 theoretical competencies divided into personal behavior, interpersonal relationships, and communication. These competencies encompass dimensions of self-awareness, self-management and motivation, social awareness and prosaically behavior, and decision-making [94]. Managers were asked to assess their own

emotions, as well as the emotions of their employees, and provide explanations for why employees experience certain emotions. The 16 items, such as "I am empathetic and considerate towards the feelings and emotions of others" and "I am self-driven and motivated" were used to measure managers' EI. The Cronbach's Alpha score for the EI construct was 0.753.

**4.1.2 Strategic human resource management.** To test the hypotheses, the following variables were used for SHRM: staffing, training, compensation, and performance appraisal. For measuring COVID-19 staffing, we adopted the approach of Kutieshat and Farmanesh [95]. However, we modified the original statements to reflect the post-COVID-19 context. To provide an illustration, specific phrases were incorporated into the survey items to reflect the impact of the post-COVID-19 environment, such as "employment and employing selection have been scaled back. In other words, it is restricted for significant positions." This particular construct comprises four items. The measurement of COVID-19 training was adapted from Kutieshat and Farmanesh's [95] work and included two items. An example item is "the organization provides adequate job training to employees in the post-COVID-19 environment." Similarly, the construct of COVID-19 compensation was derived from the research of Kutieshat and Farmanesh [95], and it comprised three items. For example, one item assessed the presence of an individual performance-based reward system in the post-COVID-19 environment. The construct for COVID-19 performance appraisal was developed by Bieńkowska et al. [36]. In the time of the post-COVID-19 environment, the construct encompasses a total of eight items. These items specifically pertain to the HR department's role in ensuring that employees are well-informed about work-related matters and have a clear understanding of their performance. We added the phrase "post-COVID-19 environment" to each question to ensure managers' attention to the specific context. This accurately reflects the situation in the post-COVID-19 era, where employees are facing new circumstances and experiencing the differential impact of SHRM [32]. The Cronbach's Alpha scores for the SHRM construct was 0.806, SHRM-Staffing was 0.705, SHRM-Training was 0.803, SHRM-Compensation was 0.703, and SHRM-Performance Appraisal was 0.719.

**4.1.3 Psychological contract.** The measurement of the PC utilized a 17-item scale originally developed by Millward and Hopkins [96] and used in previous studies [97]. To assess the PC, we adopted the scale revised by Wang et al. [97], which consists of two dimensions: transactional and relational. The transactional dimension is based on the concept of reciprocation and involves extrinsic rewards, with 10 questions. The relational dimension focuses on the mutual relationship between employees and employers and includes organizational support and development opportunities, with 7 items. In this study, we specifically considered the manager's perception of the PC. Therefore, we modified the questions accordingly. For example, one question could be "Does this employee feel like part of a team in this organization?" (Relational), or "Does this employee come to work solely to get the job done?" (Transactional). The Cronbach's alpha value for the PC scale was 0.81, indicating acceptable internal consistency.

**4.1.4 Affective organizational commitment.** The authors utilized a six-item scale, proposed by Allen and Meyer [98], to measure AOC. This scale is widely recognized and has demonstrated high reliability in previous research. To specifically capture the manager's perspective on AOC, the questions were rephrased accordingly. For instance, a sample item could be "Does this employee feel personally connected to the organization?" The Cronbach's alpha coefficient for this scale was 0.788, indicating acceptable internal consistency.

To assess internal consistency reliability, both composite reliability and Cronbach's alpha values were calculated and found to be 0.7 or higher, as shown in Table 1. The estimates of average variance extracted (AVE) were also computed and found to be 0.503 or higher, confirming convergent validity. Discriminant validity through Fronell-Larcker [99] was evaluated

**Table 1. Reliability and convergent validity.**

| Constructs | Cronbach's alpha | Composite Reliability | Average Variance Extracted (AVE) |
|---|---|---|---|
| AOC | 0.788 | 0.845 | 0.61 |
| EI | 0.753 | 0.753 | 0.503 |
| EI_DM | 0.74 | 0.747 | 0.657 |
| EI_SA | 0.705 | 0.727 | 0.527 |
| EI_SAPB | 0.706 | 0.727 | 0.528 |
| EI_SMM | 0.717 | 0.735 | 0.636 |
| PC | 0.81 | 0.81 | 0.513 |
| PC_RA | 0.769 | 0.79 | 0.519 |
| PC_TRANS | 0.809 | 0.818 | 0.511 |
| SHRM | 0.806 | 0.81 | 0.51 |
| SHRM_CO | 0.703 | 0.72 | 0.624 |
| SHRM_PA | 0.719 | 0.733 | 0.544 |
| SHRM_ST | 0.705 | 0.735 | 0.527 |
| SHRM_TR | 0.803 | 0.805 | 0.835 |

AOC: Affective organizational commitment EI-DM: (emotional intelligence-decision making); EI-SA: (emotional intelligence- self-awareness; EI-SAPB (emotional intelligence- self-awareness and prosocial behaviour); EI-SMM (emotional intelligence- self management and motivation); PC-RA psychological contract- Transactional PC-TRANS: psychological contract rational); SHRM-CO(strategic human resource management–compensation); SHRM-PA(strategic human resource management- performance appraisal); SHRM-ST(strategic human resource management- strategic staffing); SHRM-TR strategic human resource management- strategic training).

and presented in Tables 2 and 3 for all constructs. The results indicate that the square roots of each construct's AVE were greater than its correlation with other constructs, providing evidence of discriminant validity for the instrument.

## 4.2 Results

**4.2.1 Descriptive statistics.** In total, 22.73% of respondents work at public universities, while 77.27% work at private universities, which aligns with the general statistics of private universities in Georgia. The majority of respondents, accounting for 55.24%, were female. Additionally, 33.22% of respondents are up to 35 years old. Table 4 provides the nature of the sample.

Table 5 illustrates the changes in all variables. The highest mean scores were observed for the following statements:

**Table 2. Discriminant validity (Fornell–Larcker criterion).**

| | AOC | EI_DM | EI_SA | EI_SAPB | EI_SMM | PC_RA | PC_TRANS | SHRM_CO | SHRM_PA | SHRM_ST | SHRM_TR |
|---|---|---|---|---|---|---|---|---|---|---|---|
| AOC | 0.781 | | | | | | | | | | |
| EI_DM | 0.479 | 0.81 | | | | | | | | | |
| EI_SA | 0.56 | 0.417 | 0.726 | | | | | | | | |
| EI_SAPB | 0.527 | 0.449 | 0.429 | 0.726 | | | | | | | |
| EI_SMM | 0.348 | 0.349 | 0.485 | 0.412 | 0.797 | | | | | | |
| PC_RA | 0.576 | 0.335 | 0.501 | 0.526 | 0.367 | 0.721 | | | | | |
| PC_TRANS | 0.644 | 0.432 | 0.518 | 0.465 | 0.421 | 0.651 | 0.715 | | | | |
| SHRM_CO | 0.586 | 0.364 | 0.47 | 0.439 | 0.381 | 0.556 | 0.609 | 0.79 | | | |
| SHRM_PA | 0.673 | 0.444 | 0.568 | 0.452 | 0.391 | 0.575 | 0.676 | 0.615 | 0.737 | | |
| SHRM_ST | 0.556 | 0.384 | 0.444 | 0.396 | 0.358 | 0.543 | 0.573 | 0.559 | 0.715 | 0.726 | |
| SHRM_TR | 0.267 | 0.242 | 0.3 | 0.307 | 0.308 | 0.34 | 0.266 | 0.344 | 0.355 | 0.422 | 0.914 |

**Table 3. Discriminant validity of overall constructs.**

|  | AOC | EI | PC | SHRM |
|---|---|---|---|---|
| AOC | 1.000 |  |  |  |
| EI | 0.642 | 0.753 |  |  |
| PC | 0.673 | 0.657 | 0.909 |  |
| SHRM | 0.684 | 0.666 | 0.741 | 0.795 |

- "The effort which I put into my job is fairly rewarded in the post-COVID-19 environment" in the strategic performance appraisal dimension, with a mean score of 4.126.

- "I have good relationships with my workmates" in the self-awareness and prosocial behavior dimension of EI, with a mean score of 4.252.

- "This employee feels personally attached to my work organization in the post-COVID-19 environment" in the AOC dimension, with a mean score of 4.08.

- "This employee feels this company reciprocates the effort put in by its employees in the post-COVID-19 environment" in the rational dimension of the PC, with a mean score of 4.168. These findings suggest that higher education organizations are undergoing shifts in the post-COVID-19 environment, necessitating a renegotiation of the PC between SHRM practitioners and their universities. To ensure the validity of the analysis, questions with outer loadings below 0.4 were removed from the model. Additionally, Hair et al. ([100], p.103) recommend considering the removal of indicators with outer loadings between 0.4 and 0.7 only if a total of 19 questions were excluded from the analysis because their removal resulted in an improvement in composite reliability or average variance extracted, surpassing the recommended threshold. Table 5 shows the results of outer loadings.

**4.2.2 Hypotheses testing.** We run correlation analysis for hypotheses testing (see Table 6). The findings indicated a positive correlation between EI and SHRM (r = 0.600,

**Table 4. Demographic variables.**

|  |  | Frequency | Percent |
|---|---|---|---|
| Gender | Male | 128 | 44.76 |
|  | Female | 158 | 55.24 |
| Age | 26–34 | 95 | 33.22 |
|  | 35–43 | 75 | 26.22 |
|  | 44–52 | 49 | 17.13 |
|  | 53 | 67 | 23.43 |
| University | Public | 65 | 22.73 |
|  | Private | 221 | 77.27 |
| Experience | Less Than 10 Years | 46 | 16.08 |
|  | 10–20 Years | 58 | 2.28 |
|  | More Than 10 Years | 182 | 63.64 |
| Education | Diploma | 49 | 17.13 |
|  | Undergraduate | 114 | 39.86 |
|  | Postgraduate | 123 | 43.01 |
| Employment | Part Time | 34 | 11.89 |
|  | Full Time | 252 | 88.11 |
| Total |  | 286 | 100 |

**Table 5. Outer loadings.**

| Constructs | Items | Outer Loading | Minimum | Maximum | Mean | Standard Deviation | VIF |
|---|---|---|---|---|---|---|---|
| AOC | AOC1 | 0.849 | 2 | 4 | 3.983 | 0.177 | 1.889 |
| | AOC2 | 0.874 | 2 | 5 | 3.993 | 0.301 | 1.966 |
| | AOC5 | 0.610 | 1 | 5 | 4.08 | 0.379 | 1.306 |
| | AOC6 | 0.765 | 2 | 5 | 4.063 | 0.369 | 1.607 |
| EI_DM | EI_DM1 | 0.844 | 2 | 5 | 4.224 | 0.508 | 1.599 |
| | EI_DM2 | 0.800 | 2 | 5 | 4.133 | 0.438 | 1.369 |
| | EI_DM3 | 0.785 | 2 | 5 | 4.213 | 0.48 | 1.512 |
| EI_SA | EI_SA1 | 0.766 | 1 | 5 | 4.206 | 0.538 | 1.323 |
| | EI_SA2 | 0.791 | 2 | 5 | 4.234 | 0.546 | 1.446 |
| | EI_SA3 | 0.714 | 1 | 5 | 4.108 | 0.521 | 1.44 |
| | EI_SA4 | 0.621 | 2 | 5 | 4.154 | 0.52 | 1.301 |
| EI_SAPB | EI_SAPB1 | 0.642 | 1 | 5 | 4.143 | 0.506 | 1.245 |
| | EI_SAPB3 | 0.775 | 1 | 5 | 4.098 | 0.499 | 1.271 |
| | EI_SAPB4 | 0.737 | 2 | 5 | 4.252 | 0.535 | 1.408 |
| | EI_SAPB5 | 0.745 | 2 | 5 | 4.175 | 0.485 | 1.419 |
| EI_SMM | EI_SMM1 | 0.832 | 2 | 5 | 4.094 | 0.517 | 1.455 |
| | EI_SMM2 | 0.819 | 2 | 5 | 4.133 | 0.446 | 1.39 |
| | EI_SMM3 | 0.738 | 2 | 5 | 4.245 | 0.525 | 1.377 |
| PC_RA | PC_RA1 | 0.766 | 1 | 5 | 4.157 | 0.554 | 1.45 |
| | PC_RA2 | 0.697 | 2 | 5 | 4.168 | 0.515 | 1.447 |
| | PC_RA5 | 0.687 | 2 | 5 | 4.136 | 0.514 | 1.468 |
| | PC_RA6 | 0.803 | 2 | 5 | 4.087 | 0.511 | 1.706 |
| | PC_RA7 | 0.639 | 1 | 5 | 4.126 | 0.534 | 1.291 |
| PC_TRANS | PC_TRANS1 | 0.742 | 1 | 5 | 4.143 | 0.583 | 1.516 |
| | PC_TRANS3 | 0.752 | 2 | 5 | 4.098 | 0.513 | 1.572 |
| | PC_TRANS4 | 0.722 | 1 | 5 | 4.063 | 0.498 | 1.471 |
| | PC_TRANS6 | 0.672 | 1 | 5 | 4.147 | 0.528 | 1.437 |
| | PC_TRANS8 | 0.757 | 1 | 5 | 4.08 | 0.499 | 1.637 |
| | PC_TRANS9 | 0.635 | 2 | 5 | 4.196 | 0.588 | 1.354 |
| SHRM_CO | SHRM_CO1 | 0.825 | 2 | 5 | 4.017 | 0.349 | 1.343 |
| | SHRM_CO2 | 0.800 | 2 | 5 | 4.038 | 0.327 | 1.517 |
| | SHRM_CO3 | 0.743 | 2 | 5 | 4.059 | 0.41 | 1.334 |
| SHRM_PA | SHRM_PA2 | 0.730 | 2 | 5 | 4.01 | 0.318 | 1.323 |
| | SHRM_PA3 | 0.822 | 2 | 5 | 4 | 0.302 | 1.585 |
| | SHRM_PA4 | 0.719 | 2 | 5 | 3.979 | 0.354 | 1.313 |
| | SHRM_PA8 | 0.670 | 1 | 5 | 4.126 | 0.507 | 1.323 |
| SHRM_ST | SHRM_ST1 | 0.651 | 2 | 5 | 4.091 | 0.441 | 1.256 |
| | SHRM_ST2 | 0.778 | 2 | 5 | 4.038 | 0.404 | 1.43 |
| | SHRM_ST3 | 0.795 | 1 | 5 | 4.056 | 0.447 | 1.337 |
| | SHRM_ST4 | 0.669 | 2 | 5 | 4.122 | 0.445 | 1.293 |
| SHRM_TR | SHRM_TR1 | 0.909 | 2 | 5 | 4.031 | 0.328 | 1.816 |
| | SHRM_TR2 | 0.919 | 2 | 5 | 4.073 | 0.371 | 1.816 |

p < 0.01) and between SHRM and AOC (r = 0.693, p < 0.01), supporting hypothesis 1and 2. It was also confirmed that SHRM and PC were positively correlated in moderate strength (r = 0.679, p < 0.01), therefore supporting H3. The results also reveal a positive moderate relationship between PC and AOC (r = 0.650, p < 0.01), also supporting hypothesis 4.

**Table 6. Correlation matrix.**

|  | AOC | EI | PC | SHRM |
|---|---|---|---|---|
| AOC | 1 | .604** | .650** | .693** |
| EI | .604** | 1 | .576** | .600** |
| PC | .650** | .576** | 1 | .679** |
| SHRM | .693** | .600** | .679** | 1 |

**. Correlation is significant at the 0.01 level (2-tailed)

SEM with PLS method was applied for the analysis of the influence of EI on SHRM and AOC, as well as the influence of PC on AOC. Table 7 shows direct effects between variables. Fig 2 shows the results of PIS Algorithm and Fig 3 results of bootstrapping. The analysis confirmed the positive influence of EI on SHRM ($\beta = 0.666$, $p < 0.01$), SHRM on AOC ($\beta = 0.411$, $p < 0.01$) as well as a positive influence on PC ($\beta = 0.741$, $p < 0.01$). Confirming hypotheses 1, 2 and 3. In addition there was a positive impact of PC on AOC ($\beta = 0.369$, $p < 0.05$). The $R^2$ for AOC and PC were 0.548 and 0.529, which are considered as moderate impact, followed by SHRM (0.443) which is considered as week impact. Furthermore, Table 8 represents that the indirect effect of SHRM on AOC is significant. To find the role of mediator, the variance accounted for (VAF) was calculated, which determines the size of the indirect effect in relation to total effect. The VAF was 39.91%, which shows that the PC has the partial mediation role in the relationship between SHRM and AOC. Based on Table 9, the model has a good fit (SRMR = 0.089, NFI = 0.790, $\chi^2$ = 327.937).

## 5. Discussion

COVID-19 has undoubtedly resulted in the emergence of a complex and challenging environment for SHRM practitioners who needed to come up with creative solutions to sustain their organizations and assist their staff in coping with the difficulties of this unheard-of situation [49]. The primary objective of this study was to investigate the role of SHRM practitioners' EI on AOC in higher education in Georgia. The research conducted for the purpose of this study showed that EI has a positive influence on SHRM during post-COVID-19 and in turn has a positive effect on the AOC in the higher education (Hypothesis 1 and 2). Regarding this matter, it is crucial to emphasize the dearth of comparable research conducted within this particular context. However, a general literature review conducted by Hamouche [49] during COVID-19, emphasized the current situation has provided SHRM practitioners with a unique opportunity to meet the strategic objectives of the organization to overcome the silent impact of COVID-19. By integrating the organizational focus on emotions with the value of SHRM, we can assert that EI competencies have the potential to reshape SHRM practitioners into a group of individuals who prioritize people-oriented activities. These activities aim to enhance

**Table 7. Direct effects.**

| Hypotheses | Path | Path Coefficient | SD | Mean | T-value | P-value | Results |
|---|---|---|---|---|---|---|---|
| H1 | EI -> SHRM | 0.666*** | 0.102 | 0.65 | 6.530 | 0.000 | Accept |
| H2 | SHRM -> AOC | 0.411*** | 0.142 | 0.432 | 2.902 | 0.004 | Accept |
| H3 | SHRM -> PC | 0.741*** | 0.073 | 0.729 | 10.083 | 0.000 | Accept |
| H4 | PC -> AOC | 0.369** | 0.175 | 0.327 | 2.106 | 0.035 | Accept |

***$p < .01$, **$p < .05$

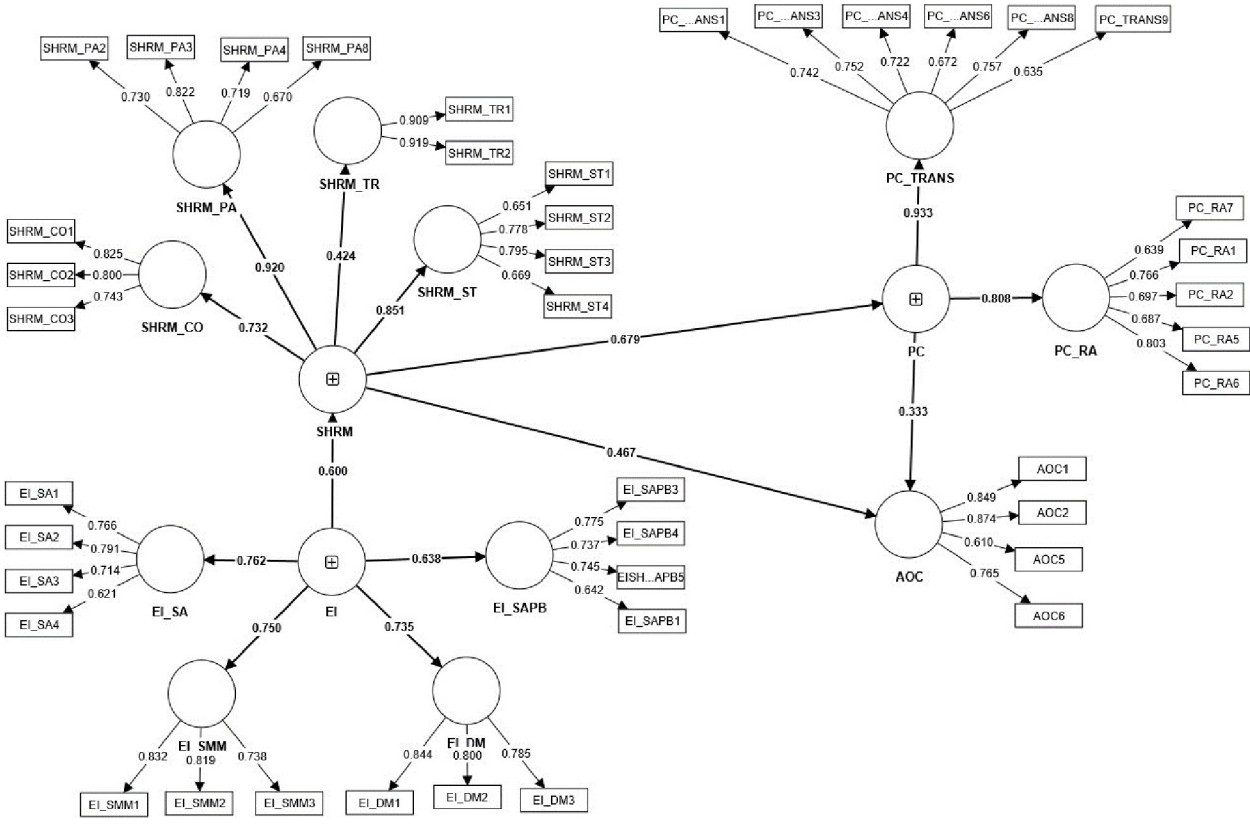

**Fig 2. Results of PLS algorithm.**

the efficient acquisition, utilization, and retention of employees within the organization. As we noted before, a post-Covid-19 environment demands greater emotional intelligence [101]. Moreover, providing guidance on how to keep highly committed employees in Georgia's higher education institutions during a crisis and after an unexpected transformations. This recommendation is in line with other recent literature in the field of education management

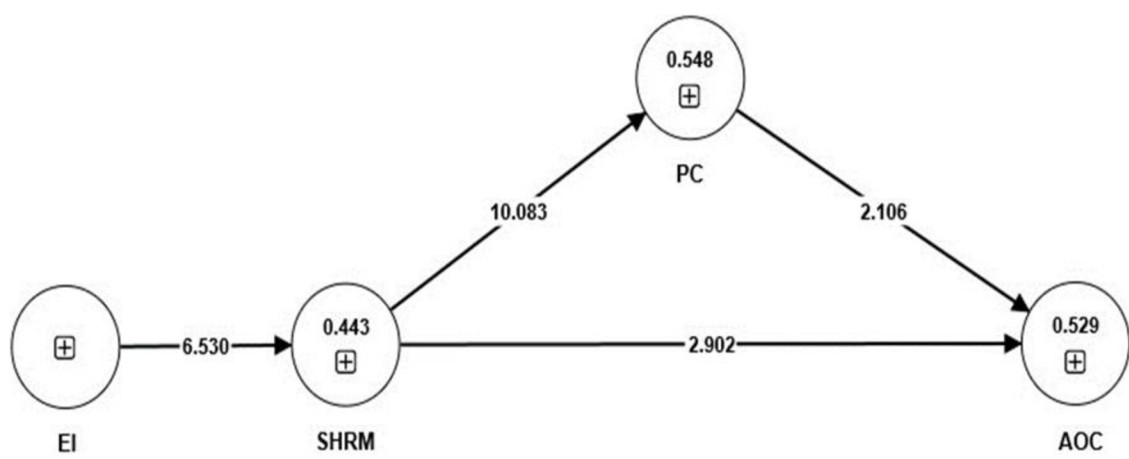

**Fig 3. Results of bootstrapping.**

**Table 8. Indirect effects.**

| Hypotheses | Path | Path Coefficient | SD | Mean | T-value | P-value | Results |
|---|---|---|---|---|---|---|---|
| H5 | SHRM -> PC -> AOC | 0.273** | 0.134 | 0.242 | 2.041 | 0.041 | Accept |

***p < .01, **p < .05

[102]. For instance, Sharma and Ramesh [103] show that to achieve organizational goals in the post-COVID-19 pandemic environment, strategic thinking in HRM will be required. Furthermore, our research indicates that organizations in higher education can enhance commitment and sense of belonging by developing their EI. This is particularly crucial in the aftermath of the COVID-19 crisis, and we propose using a modified approach to address the challenges posed by the pandemic [63].

As noted, SHRM practitioners play a vital role in addressing staffing challenges, such as employment classifications and work arrangements in post-COVID-19. This is in line with Vaccaro et al. [104] and Green [38] which believed that COVID-19 will likely lead to a significant reduction in staffing of academic library during the next three to five years.

Regarding compensation and benefits, which are related to employee motivation after COVID-19, the results of the current article are consistent with the previous study of Mabaso and Dlamini [105] and Shtembari et al. [44], who argued the need to redefine compensation and benefits after COVID-19 and reported that higher education institutions must strengthen their remuneration strategies in order to increase employee commitment and efficiently provide excellent results.

Strategic training, the other SHRM practices in the post-COVID-19 period, also harmonized with the previous study conducted by Thilagaraj and Rengaraj [41], emphasized retraining and showed that the post-COVID-19 workplace would need different skills and technologies than were required before COVID-19. According to a recent study conducted by Camilleri [106], higher education needs to use interactive technologies, in a post-COVID-19 era, and invest in online learning infrastructures. Additionally, this research tested the SHRM practitioner's EI and PC in the higher education in post- COVID-19 (Hypothesis 3). This proposed hypothesis was supported. The findings revealed that PC is a key psychological pathway for SHRM practitioners to employ effective strategies in staffing, training, compensation, and performance appraisal to influence employee attitudes and behaviors. These findings support the results presented by Mihalache and Mihalache [78], Kowal et al. [80] and Wang et al. [97] and contradict the perspective of Ronnie et al. [56] who argued that psychological contracts have shifted or been breached in the COVID -19 outbreak in higher education institutions. The organization's decision to support employees during the COVID-19 outbreak has a major impact on employee feeling and reciprocal relationship and building positive emotions in the workplace is necessary for managers to understand and ensure employee job-related well being [97]. They argue that managers with a low level of emotional competence are unable to

**Table 9. Model fit.**

| | Saturated model | Estimated model |
|---|---|---|
| SRMR | 0.073 | 0.089 |
| d_ULS | 0.354 | 0.526 |
| d_G | 0.168 | 0.204 |
| Chi-square | 289.494 | 327.937 |
| NFI | 0.815 | 0.790 |

recognize the needs of employees and motivate them in organizations. It means due to lack of emotional competencies they fail to perform effective HRM practices to fulfil employees needs and are not able to develop employees or provide an opportunity for retaining, which in turn leads to negative attitude like turnover intention and negative behaviours e.g., employee turnover. In other words, the lack of emotional competencies affects the lack of PC satisfaction. The importance of this issue is very crucial during this time of uncertainty and crisis due to the management of employees' emotions as a result of a good feeling about working conditions.

According to survey results, PC is positively related to AOC in the higher education in post- COVID-19 (Hypothesis 4). This research findings are in line with research done during the COVID-19 outbreak and post-pandemic environment. For example, Yu [57] assume that PC fulfillment has a positive impact on intrinsic motivation, and affective commitment among 405 employees working remotely. In addition, our findings are consistent with those made by Ronnie et al. [56], who acknowledged the breach and violation of PC and suggested that in the post-COVID-19 higher education scenario, PC can be clarified and renegotiated between academics and their institutions. Universities need to be aware of how they might influence the academic-institution relationship through leadership. As it has been demonstrated to favorably influence commitment, if the psychological contract that supports this connection is correctly maintained, this is likely to have favorable consequences for all parties.

Furthermore, the results show that SHRM practitioner's EI and PC in post-COVID-19 can improve employees' affective commitment to the organization (H5). The correlations between PC and AOC revealed in this research are in line with those previously conducted by Karani et al. [86] and Ababneh [84] who emphasized that PC fulfilment has a significant impact on employees' emotional attachment during the COVID-19 crisis. As discussed before, SEM results confirmed the direct influence of SHRM dimensions and indirect on AOC mediated by PC. Therefore, H5 is confirmed partly. Looking at previous studies by Kalyani [92] it could be stated that SHRM practitioners in the post epidemic era will enhance PC which in turn will lead to the high employee's commitment. It means AOC increases not only by SHRM, but also by PC, as it was found in this study.

## 6. Conclusion

### 6.1 Theoretical implications

The results of this study contribute to our understanding of the connections between SHRM practitioners' EI, PC, and AOC in the post-COVID-19. This study is useful as it is applicable to changing the context demonstrated in the COVID-19 pandemic. They fill a research gap identified in a general literature review by Hamouche [49] concerning strategic thinking in HRM during COVID-19. This study adds to the knowledge of resource-based theory which gives a new perspective to SHRM practitioners to implement new measures in SHRM practices, such as staffing, training, compensation, performance appraisal and social exchange theory which are related to many organizational outcomes. At the moment there are just few studies (see: Neuwirth et al. [107]; Ewing [108]; Camilleri [106]) published in post-COVID-19 environment in educational sector. However, they more focus on well-being and service quality. This study particular focuses not on educators but on SHRM practitioners of educational sector and explores their EI, PC and AOC.

The results of this study provide a number of contributions. Firstly, the study contributes to the broader application of social exchange theory to psychological contracts explaining relationship and interaction between employees and employers. Secondly, results revealed EI has a positive impact on every practice of SHRM practitioners in higher education. This finding adds to recent findings of Lee et al. [101] who identified that emotions are important

competence for managers and lead to better understanding of employees' behavior outcomes. The educational sector can achieve this by effectively implementing these strategies. This is in line with Kutieshe and Farmanesh [95] suggestions. Therefore, this study has taken different perspectives, focusing on the managers and not the employees. Moreover, the EI of managers utilizes human resources to serve the employees' strategic needs. This element of the presented findings is an original contribution to expanding the better understanding of the role of SHRM practitioners' EI in higher education. Fourthly, the incorporation of EI as an antecedent of SHRM provides guidance for managers in higher education on which SHRM practices are the most important for them to be more successful in their roles and for improving AOC.

## 6.2 Managerial and practical implications

The highlighted results could summarize when employee's strategic needs in terms of PC fulfilled, as well as how it may help employees attached and committed to the organization emotionally. To consider employee's needs, it is important to pay attention and adapt government plans and regulations to the organizational context. Since the COVID-19 outbreak has altered organizations and most of them overwhelmed by challenges resulting from COVID-19 (as previously pointed out by Hamouche [49], e.g., lack of trust and tension between employees and employers, it is crucial to ensure that employee' needs is aligned with the universities' strategic goals. Sustaining communication within an organization helps to reduce stress and increase trust [49, 109]. Not only that, but it is also crucial for higher education to communicate relevant data related to the organization's strategic direction to their employees and transform SHRM practices into a set of people-oriented activities that promote the effective procurement and retention of employees and provide feedback to them, which will help higher educations to retain their human capital and avoid employee's turnover in a time of crisis and after an unprecedented situation. Furthermore, SHRM practitioners should play a strategic role by supporting and employees' training on how to overcome the difficulties and to cope with working challenges after COVID-19 related to employees' development [110]. According to the necessity to develop lacking job security, special training could be proposed by taking strategic approach for employees. Training programs for EI development of SHRM practitioners is necessary for fostering employee's commitment and their retention. In addition, information technology is crucial for rethinking HRM strategies for proposing new models of managing human resources to ensure sustainability of organizations. Additionally, it offers higher education organizations the opportunity to optimize traditional HR practices and improve workplace planning, foster job control and costs of their resources. Furthermore, it is necessary to provide PC, supportive environment, and cohesive culture through consultation systems for employees. Consultation systems support employees' connections and interactions and all together could have a positive impact on AOC.

## 6.3 Limitations and further research directions

Firstly, the study design employed in this research was cross-sectional. To further explore how AOC changes during the post-COVID-19 period and the return to normal, it is recommended to conduct a longitudinal study. This would provide insights into how AOC progresses over time for SHRM practitioners with high or low EI. Next, a convenient sampling was used for this study, with participants selected from private universities in Tbilisi. However, self-report data are subject to sampling bias. Therefore, to reduce the likelihood of common method bias in self-reported studies in future research, the authors suggest collecting data from multiple sources, such as educators survey, document (content) analysis, focus groups. Thirdly, the study did not explore employees' perceptions and experiences during the post-COVID 19 era,

nor the role of SHRM in this context. The focus of this study was on SHRM practitioners rather than educators, which differs from most SHRM studies. However, future research could gather data from educators and compare the results.

Fourthly, while this study did not compare changes in EI post-COVID-19, it was not the main focus of the research. This could be expanded in a future study. Fifthly, future studies should consider examining other dimensions of OC, such as continuance and normative commitment, to address additional questions that were not answered in this study. Next, this research did not consider the EI of managers at different levels. The EI of middle managers or line managers at various hierarchical levels can influence the implementation of HRM practices and AOC. Therefore, it is recommended that future studies measure the EI of management in large organizations. Finally, this study focused solely on one country. To broaden the scope, future research should explore other countries and compare the results in relation to demographic characteristics. It is suggested that future studies examine the relationship between OC and post-COVID-19, how SHRM practitioners' EI influences the OC in different countries, how SHRM influences OC, and the other factors of social exchange theory that are related to OC in different countries. Additionally, the influence of EI on PC and OC should be investigated.

## 6.4 Conclusion

The findings from the current study highlight the importance of EI among SHRM practitioners in university management during the post-COVID-19 period in Georgia. It is worth noting that this research was conducted at a crucial time when continuous development and commitment were identified as key challenges for the future development of Georgian educational institutions, with the aim of aligning them with European higher education standards. The adoption of new legislation from different countries can provide support to higher education organizations in times of crises and unexpected transformations. The results of the study indicate that SHRM can enhance AOC. Furthermore, SHRM, with its four dimensions, indirectly influences AOC through its mediation by PC. Additionally, the findings demonstrate that EI skills are essential for managers to possess in order to develop effective strategies. SHRM practitioners with high EI are more likely to prioritize the long-term needs of employees, such as creating motivating working conditions and providing opportunities for professional development. It is important to recognize that the higher education landscape is undergoing significant changes in the post-COVID-19 environment, which necessitates the renegotiation of the PC between employees and their universities. Furthermore, maintaining positive relationships within the workplace is crucial for fostering a sense of self-worth, belonging, and community, which ultimately leads to higher levels of AOC.

## Supporting information

**S1 Appendix. Methods and sampling.**
(DOCX)

**S2 Appendix. Variables and measures.**
(DOCX)

## Author Contributions

**Conceptualization:** Roya Anvari.

**Data curation:** Roya Anvari, Rokhsareh Mobarhan.

**Formal analysis:** Mariam Janjaria.

**Methodology:** Roya Anvari.

**Writing – original draft:** Roya Anvari.

**Writing – review & editing:** Roya Anvari, Vilmantė Kumpikaitė-Valiūnienė, Siavash Hossein-pour Chermahini.

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
