## [Decision Letter · Decision Letter 0]

25 Sep 2023

PONE-D-23-26406strategic human resource management practitioners’ emotional intelligence and affective organizational commitment in higher education institutions in Georgia during post-COVID-19PLOS ONE

Dear Dr. Anvari,

Thank you for submitting your manuscript to PLOS ONE. After careful consideration, we feel that it has merit but does not fully meet PLOS ONE’s publication criteria as it currently stands. Therefore, we invite you to submit a revised version of the manuscript that addresses the points raised during the review process.

We look forward to receiving your revised manuscript.

Kind regards,

Rogis Baker, Ph.D

Academic Editor

PLOS ONE

Journal Requirements:

   "This research received no external funding. "

5. Please ensure that you include a title page within your main document. You should list all authors and all affiliations as per our author instructions and clearly indicate the corresponding author.

Reviewers' comments:

Reviewer's Responses to Questions

**Comments to the Author**

1. Is the manuscript technically sound, and do the data support the conclusions?

Reviewer #1: Yes

Reviewer #2: Partly

Reviewer #3: Yes

Reviewer #4: Yes

Reviewer #5: Yes

2. Has the statistical analysis been performed appropriately and rigorously? 

Reviewer #1: Yes

Reviewer #2: Yes

Reviewer #3: No

Reviewer #4: Yes

Reviewer #5: Yes

3. Have the authors made all data underlying the findings in their manuscript fully available?

Reviewer #1: Yes

Reviewer #2: Yes

Reviewer #3: Yes

Reviewer #4: Yes

Reviewer #5: No

4. Is the manuscript presented in an intelligible fashion and written in standard English?

Reviewer #1: Yes

Reviewer #2: No

Reviewer #3: Yes

Reviewer #4: Yes

Reviewer #5: Yes

5. Review Comments to the Author

Reviewer #1: Although, author has found positive correlations between EI and SHRM (hypothesis 1) and SHRM and PC (hypothesis 3) mentioned at line number 561 and 563, but these correlations are not very strong. Therefore reasons for the same may be explained.

Reviewer #2: The idea is good and there is a lot of potential in the topic. However, the writing needs improvement in terms of sentence construction, flow and consistency in Abstract, Introduction and Discussion. Also, there are some mistakes which need to be removed.

Abstract and Introduction

The rationale of the study needs to be refined. It is not clear why the study needs to be done from the strategic HRM professionals’ perspective in the higher education domain. Sentences need to be reworded (e.g line 13 to 16, line 35-38), some vocabulary needs to be modified (e.g. procure in line 20) and consistency in arguments need to be revisited (Line 63-65).

Theory and Hypothesis

Literature on EI and SHRM is well covered. However, The SHRM variable needs to be conceptualized more clearly, specifically in terms of literature indicating its significance in the higher education domain. Also, OC can be elaborated further in terms of the three components.

Arguments for hypotheses need to be more clear and detailed, and with direct reference to higher education sector.

Data Analysis and Discussion

Data analysis is strong

Table 2 needs to be labelled appropriately. The table is giving information about Discriminant Validity but doesn’t contain any values representing it.

The arguments in discussion need to be more convincing about the results being specific to the context of Covid-19 situation (contribution stated) and contribution to the higher education domain.

General Comments

There seem to be some errors. In some places in analysis, SHRM is mentioned as an independent variable (Section 4.1.2). Also, the paper mentions POS at the end, which is not a variable under study. AOC and OC seem to have been used interchangeably through all the sections of the paper. In some places, the paper is talking about PC of employees, and in other places PC of SHRM practitioners.

Reviewer #3: The topic under consideration is intriguing and warrants significant attention from both scholars and practitioners within the realm of higher education, especially following the COVID-19 endemic. The study commences with a compelling introduction that sets a strong foundation. Overall, the research problem addressed in this paper holds substantial practical and academic significance. However, there are some notable issues that require attention. My recommendation for the current version of the paper is to undergo minor revisions. The authors are encouraged to address and provide responses to the following concerns.

1. Throughout the manuscript, there are instances where the author has employed subjective language, notably the use of "We." It is advisable for the author to carefully review these instances and consider removing such subjective language to enhance the paper's objectivity.

2. In the theoretical framework, it is essential to establish a more explicit connection to recent research within the higher education sector, particularly in the context of the post-COVID-19 environment. The author has made efforts to engage in discussions across various sectors; however, it is noticeable that the higher education sector is conspicuously absent.

3. The author has repeatedly emphasized the need for SHRM practitioners to address various questions to tackle the challenges outlined. However, there is ambiguity regarding whether all of these questions directly contribute to the outcomes presented in the paper. Furthermore, in the discussion section, these questions are not adequately addressed in the context of the study. It is essential to provide clarity and reframe these questions within the specific context of the research.

4. Upon utilizing the provided outer loading, I randomly attempted to compute the Composite Reliability (CR), Average Variance Extracted (AVE), and the square root of AVE. However, it has come to my attention that there is a significant disparity in the calculated values. Here are a few examples for clarity:

• For the AOC Construct: CR = 0.780, AVE = 0.47649075, and the square root of AVE = 0.690283094.

• For the SHRM_ST Construct: CR = 0.724, AVE = 0.469198, and the square root of AVE = 0.684980292.

These calculations indicate a discrepancy that warrants attention and clarification. Please re-analyze and calculate the entire construct to avoid such inconsistency.

Additionally, it's important to assess reliability, convergent validity and discriminant validity for the overall construct to ensure that they are measuring distinct aspects of the construct and are related to other constructs in the expected ways.

Reviewer #4: 1. Introduction: main points are mentioned, but there are some discrepancies.

(1) Research objective has some ambiguity. research objective should not be confused with implications. research objective is more about the relationship between the two and the many, rather than about implications. It can be changed to "Being a strong sense of commitment among employees can positively promote the organisation's future success. "

(2) Unclear definition of research gap. There is no talk of research gaps in which niche.

(3) Try to avoid the use of first person.

2. The literature review is not adequate and there are numerous omissions.

(1) The Theoretical framework needs further definition of relevant involved concepts or variables, i.e., you need to use the definition of concepts in this paper. For example, "In this study, EI is defined as..."

(2) Some definitional errors. For example, "Strategic Human Resource Management (first introduced in the 1990s) "

Reference:

[1]Hartel, C. E. J., & Fujimoto, Y. (2014). Human Resource Management (3rd ed.). Pearson Education.

[2]Roehl, M. T. (2019). The impact of SHRM on the psychological contract of employees: A typology and research agenda. Personnel Review, 48(6), 1580–1595. https://doi.org/10.1108/pr-02-2018-0063

[3]Wright, P. M., & Steinbach, A. L. (2022). Pivoting after almost 50 years of SHRM research: toward a stakeholder view. Asia Pacific Journal of Human Resources, 60(1), 22–40. https://doi.org/10.1111/1744-7941.12308

(3) Many parts lack relevant literature review and do not do a good work on literature search.

(i) In the section of SHRM, there are many opinions and definitions, but there is a lack of relevant theoretical support.

(ii) In part 2.2.3, how to find out the form of "Compensation"? Some literature review and summarisation is needed;

(4) Inappropriate placement of the question. Try to avoid using a lot of questions in section 2.2.3.

3. Overall, there are some research contributions, but not much. A large number of articles have been published to study the relationship between the elements of SHRM, EI, PC and so on, while the research contribution of this article focuses on the specific practice field or industrial, and the contribution to the related theories is not much. Further contributions to relevant theories need to be strengthened if further publications are required.

Reviewer #5: Dear Author

Thank you for submitting your paper to this journal.

I read your paper and gave my concern down here:

1. It seems that the title is too long and thus, the reader or audience might loose their focuses while reading. I am recommending to shorten it.

2. The abstract is not exhaustive. I will suggest to simplify and follow IMRaD concept to present it.

3. The research problem is not adequately justified.

4. Would you add CMB issue as self-reports are there.

Wish you all the best.

6. PLOS authors have the option to publish the peer review history of their article (what does this mean?). If published, this will include your full peer review and any attached files.

Reviewer #1: No

Reviewer #2: **Yes: **Smita Chaudhry

Reviewer #3: **Yes: **Hasanuzzaman Tushar

Reviewer #4: No

Reviewer #5: No

---

## [Author Response · Author response to Decision Letter 0]

8 Nov 2023

Dear Editor, 

The manuscript has been revised according to the suggestions and comments of the reviewers. 

Please kindly note that a new analysis has been added and the major revised parts are highlighted in yellow color for your convenience of re-reviewing. I hope that it's now suitable for publication. We are thankful to the reviewer for the time and effort spent on reviewing our paper. We believe that these comments help us reduce the possible confusion in the text with respect to paper’s novelty and rigor that appear to have arisen due to lack of sufficient clarity in the original version. Kindly find the attache file. 

Best regards, 

Roya Anvari

---

## [Editor Report · Decision Letter 1]

15 Nov 2023

strategic human resource management practitioners’ emotional intelligence and affective organizational commitment in higher education institutions in Georgia during post-COVID-19

PONE-D-23-26406R1

Dear Dr. Roya Anvari

We’re pleased to inform you that your manuscript has been judged scientifically suitable for publication and will be formally accepted for publication once it meets all outstanding technical requirements.

Kind regards,

Rogis Baker, Ph.D

Academic Editor

PLOS ONE
---

## [Editor Report · Acceptance letter]

14 Dec 2023

PONE-D-23-26406R1 

PLOS ONE

Dear Dr. Anvari, 

I'm pleased to inform you that your manuscript has been deemed suitable for publication in PLOS ONE. Congratulations! Your manuscript is now being handed over to our production team.

Kind regards, 

on behalf of

Dr. Rogis Baker 

Academic Editor

PLOS ONE